# Segmentation of the Myocardium on Late-Gadolinium Enhanced MRI based on 2.5 D Residual Squeeze and Excitation Deep Learning Model

**Abdul Qayyum**[1]                                 ABDUL.QAYYUM@U-BOURGOGNE.FR
**Alain Lalande**[1,2]                               ALAIN.LALANDE@U-BOURGOGNE.FR
**Thomas Decourselle**[3]                           TDECOURSELLE@CASIS.FR
**Thibaut Pommier**[2]                              THIBAUT.POMMER@CHU-DIJON.FR
**Alexandre Cochet**[1,2]                           ALEXANDRE.COCHET@U-BOURGOGNE.FR
**Fabrice Meriaudeau**[1]                           FABRICE.MERIAUDEAU@U-BOURGOGNE.FR

[1] *ImViA laboratory, University of Bourgogne Franche-Comté, Dijon, France*

[2] *Medical imaging department, University Hospital of Dijon, Dijon, France*

[3] *CASIS company (Quetigny-France)*

## Abstract

Cardiac left ventricular (LV) segmentation from short-axis MRI acquired 10 minutes after the injection of a contrast agent (LGE-MRI) is a necessary step in the processing allowing the identification and diagnosis of cardiac diseases such as myocardial infarction. However, this segmentation is challenging due to high variability across subjects and the potential lack of contrast between structures. Then, the main objective of this work is to develop an accurate automatic segmentation method based on deep learning models for the myocardial borders on LGE-MRI. To this end, 2.5 D residual neural network integrated with a squeeze and excitation blocks in encoder side with specialized convolutional has been proposed. Late fusion has been used to merge the output of the best trained proposed models from a different set of hyperparameters. A total number of 320 exams (with a mean number of 6 slices per exam) were used for training and 28 exams used for testing. The performance analysis of the proposed ensemble model in the basal and middle slices was similar as compared to intra-observer study and slightly lower at apical slices. The overall Dice score was 82.01% by our proposed method as compared to Dice score of 83.22% obtained from the intra observer study. The proposed model could be used for the automatic segmentation of myocardial border that is a very important step for accurate quantification of no-reflow, myocardial infarction, myocarditis, and hypertrophic cardiomyopathy, among others.

**Keywords:** LGE-MRI, segmentation, deep learning, 2.5 D modeling

## 1. Introduction

Late gadolinium enhancement (LGE) MRI is the cornerstone of myocardial tissue characterization, representing the most accurate and highest resolution method for myocardial infarction (MI) and non-ischemic cardiomyopathy diagnosis. In particular, phase sensitive inversion recovery (PSIR) is an efficient sequence for myocardial viability assessment(Oksuz et al., 2017). Automatic segmentation of LV myocardium based on LGE MR images is a challenging task due to the large shape variation of the heart, similarity in the signal between

myocardial scar and its adjacent blood pool, or in the signal between normal myocardium and surrounding structures and diverse intensity distributions between images. However, as both data and computational resources are increasing, fully automated and accurate segmentation of LV myocardium for LGE MR images would be possible and become a highly desirable task. Recently, deep learning-based models have been applied for LGE MRI segmentation in various disease areas. Zabihollahy et al.(Zabihollahy et al., 2019) proposed a 2D Unet to segment myocardial boundaries from 3D LGE MR images. Moccia et al.(Sara et al., 2019) proposed modified ENet based on fully-convolutional neural networks (FCNNs) on cardiac magnetic resonance with late gadolinium enhancement (CMR-LGE) images for scar segmentation and produced optimal results. Lei et al.(Lei et al., 2020) introduced Graph Net approach with multi-scale convolutional neural network (MS-CNN) for assessment of atrial scar in LGE-MRI dataset and produced good results. In this article, our focus is onto the segmentation myocardial border using LGE MRI dataset. We propose a novel 2.5 D based segmentation model for segmentation of LV myocardium on LGE MRI dataset. The main contributions in our proposed deep learning model are (i) a residual module and SE block in encoder side and introduction of special convolutional layer after each residual block, (ii) three consecutive input slices are concatenated to make a 2.5 D model, (iii) the optimization of the proposed using various hyperparameters and the final segmentation is based on a late fusion scheme.

## 2. Material and Method

### 2.1. Datasets

Three hundred forty-eight (348) late-gadolinium enhanced MRI exams have been recruited. A total number of 980 of slices (320 cases) were used for training and 28 cases (164 slices) were used for testing. For all patients, a short-axis stack of cardiac images covering the whole left ventricle from the base to the apex were acquired using MRI devices with magnetic fields of 1.5T or 3T (Siemens Healthineers, Erlangen, Germany). Gadolinium-based contrast agent (Dotarem, Guerbet, France) was administered to the patients between 8 to 10 minutes before the acquisition. Along with the raw data, the contours of the myocardium were delineated by an expert with more than 15 years of expertise in the field. Pixel sizes differed among scans between 1.25 mm x 1.25 mm to 1.91 mm x 1.91 mm and size of input image was 224x224. Intra-observer and inter-observer annotations were provided for the 28 test cases. For the manual contouring, the experts used the QIR software (https://www.casis.fr/), a software dedicated to the automatic processing of cardiovascular MRI. This software allows to manually draw the cardiac contours in a dedicated GUI. Two experts did the manual segmentation. The first one is a biophysicist with 15 years if experience, and the second one is a cardiologist of 10 years of experience in cardiology and MRI ( for the intra-observer study, these annotations were done with a time gap of several weeks and in a different order).

### 2.2. 2.5 D proposed RSE-Net Model

We have processed input images to our proposed models like an RGB color channels and stacked three consecutive slices to make 2.5 D model instead of processing the 3D volume.

The proposed model consists of a number of residuals blocks with identity function incorporated with SE module (Roy et al., 2018)and special convolutional module. We have constructed encoder block based on pre-trained ResNet50 model as a base model and applied SE block after each residual block. The squeeze & excitation (SE) block has been used at encoder side after each Residual Block. The main advantage of SE block is to extract spatial dependency (or channel dependency, or both) and reinject ('excite') this information in the feature maps. In our case SE block, first 'squeezes' along the spatial domain and 'excites' or reweights along the channels and this block is denoted as channel-wise SE (cSE) block for 2D segmentation. Features are extracted from every residual block of ResNet 50 model as a base network. 2D-SE blocks are added after each residual block in the encoder side. We have introduced specialized convolutional layer with fixed number of feature maps (k=16) after each SE blocks from encoder. Special layers are resized to the original image size through deconvolutional layers, then concatenated and passed to the last convolutional layer with 1x1 kernel size that combines linearly the fine to coarse feature maps to produce the final segmentation result. The main advantage of specialized layer is that, this layer extracts and carries the information from coarse to fine feature maps that are useful in segmentation map. We connected that specialized convolutional layer to the final layer of each block for better segmentation map and that also adds supervision at multiple internal layers of the network. The final segmentation is based on a late fusion (max voting of three models) at the pixel level. The proposed model has been implemented using Pytorch deep learning library. Adam optimizer with different learning rates and batch sizes have been used for training the models. The best learning rate of 0.0002 with Adam optimizer, a batch size of 12 and a Dice loss were used for training the proposed model. A Tesla V 100 machine containing four embedded GPUs has been used for training the proposed model.

## 3. Results and Discussion

The segmentation map generated by the model was compared with the manual segmentations thanks to performance metrics such as Dice similarity coefficients (DSC) and Hausdorff distance (HD) in 2D. The results are shown in Table 1 for base, middle and apex slices based on the proposed model and compared with the inter-observer and intra-observer studies. The proposed model produced results very close to the intra-observer variation, even a little bit more better than the inter-observer variation and outperformed existing state-of-the-art deep learning models such as simple Unet(Ronneberger et al., 2015), SegNet(Badrinarayanan et al., 2017), and Fractalnet(Yu et al., 2016) that produced overall DSC 76.93%,76.23% and 75.70% respectively. The Wilcoxon signed rank test was applied to investigate the statistically significant differences between predicted segmentation mask and manual one. The computed p-values were significant ($p<0.0001$) corresponding to a significant difference in Unet, SegNet and Fractalnet methods but not for our method ($p=0.123$). The correlation coefficient value was computed between the surface area of ground truth and predicted segmentation masks and we took the average value for all slices. Our approach achieved the best correlation values (r) between surface area from ground truth and predicted segmentation masks and lowest Bland-Altman (BA) biases (Table 1)

The main challenge is the similarity in signal, for example between myocardial scar and its adjacent blood pool, and diverse intensity distributions of images of myocardium

Table 1: The performance analysis of proposed model compared with intra observer and inter observer studies on test cases. Mean (standard deviation) for BA bias..

| Algorithms | | DSC (%) | HD (mm) | r | BA Bias (%) |
|---|---|---|---|---|---|
| Intra Observer | Base | 86.66 | 3.01 | 0.976 | 0.10(0.50) |
| | Middle | 85.24 | 2.94 | 0.961 | -0.025(0.33) |
| | Apex | 77.51 | 2.98 | 0.941 | 0.30(1.38) |
| | Overall | 83.22 | 3.26 | 0.957 | 0.11 (0.85) |
| Inter Observer | Base | 82.54 | 4.03 | 0.957 | 0.34(0.92) |
| | Middle | 81.22 | 3.87 | 0.955 | 0.18(0.73) |
| | Apex | 74.12 | 3.87 | 0.924 | 0.53(1.95) |
| | Overall | 79.25 | 4.12 | 0.945 | 0.33 (1.31) |
| Our Method | Base | 86.55 | 3.13 | 0.969 | -0.16(0.57) |
| | Middle | 84.77 | 3.65 | 0.955 | 0.30(0.87) |
| | Apex | 76.85 | 3.69 | 0.93 | 0.31(1.56) |
| | Overall | 82.01 | 3.67 | 0.959 | 0.19 (1.07) |

across different slices, particularly at basal and apical locations. Moreover, the myocardial anatomy is dissimilar between patients due to a varying severity of disease. However, the dataset is quite representative of the various cases and enables to train a reliable model. The presence of scar tissue within myocardial tissue could produce false negatives in segmentation. In order to address this issue, coarse to fine feature extraction strategy with deep network layers in residual module and SE module with specialized convolutional layer has been used to learn more feature information and improve myocardium segmentation accuracy.

## 4. Conclusion

In this paper, we have proposed a novel, fully automated ensemble model with 2.5 D strategy for myocardium border segmentation on LGE-MRI images. The proposed ensemble method shows excellent results as compared to existing state-of-the art deep learning models and lies within the intra- and inter- observer variabilities. Late fusion is providing good results but other fusion schemes are currently being investigated.

## Acknowledgments

This work was supported by the French "Investissements d'Avenir" program, ISITE-BFC project (number ANR-15-IDEX-0003)

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
