# OpenReview forum: "Segmentation of the Myocardium on Late-Gadolinium Enhanced MRI based on 2.5 D Residual Squeeze and Excitation Deep Learning Model "
_MIDL.io/2020/Conference — MIDL 2020_

### Official Review · AnonReviewer4 · 2020-02-26
**The contribution of this work is not clear, and the results are not convincing.**

**Rating:** 2
**Confidence:** 5

**Review:**

The paper proposed a new 2.5 D residual neural network for myocardial segmentation from LGE MRI. The results showed their method achieved similar accuracy to the intra-observer variation, and better than the inter-observer variation.

*Strengths
None.

*Weaknesses
1.	Regarding to the methodology in this work, what and how the network is composed of and trained. A graph illustrating the architecture is very useful.
2.	In the Sec 2.2, the paper reads, “SE module and special convolutional module” were used in the network. Why this should be used? What is the rationale or advantage of using it?
3.	In the Introduction, only one paper about segmentation of LGE MR images is referred. The authors should be aware that recently there were a LGE MRI segmentation challenge (MS-CMRseg) and cardiac MRI segmentation challenge (ACDC) from MICCAI, and ample literature is available for comparisons.
4.	Results showed they achieved Housdorff Distance (HD) of about 3-4 mm in apex and basal slices, which is very contradictory to existing literature. With better Dice scores, MS-CMRseg challenge reported the best myocardium segmentation from over 20 submissions had over 10 mm HD on LGE MRI, and ACDC reported best HD of about 5-10 mm on BSSFP MRI.

---

### Official Review · AnonReviewer2 · 2020-03-03
**The manuscript “Segmentation of the Myocardium on Late-Gadolinium Enhanced MRI based on 2.5 D Residual Squeeze and Excitation Deep Learning Model” presents a method for myocardium segmentation from CMR-LGE volumes.**

**Rating:** 2
**Confidence:** 5

**Review:**

The manuscript addresses a relevant task but presents some weaknesses. Some methodological choices are not clearly explained: for example, which are the benefits of including squeeze and excitation blocks for the addressed task?  The survey of the state of the art can be improved, e.g., by citing and discussing more relevant literature. I strongly suggest the authors to ask a native English speaker to proofread the manuscript. The overall manuscript readability also can be improved.

My specific comments can be found hereafter.

Abstract
- The authors should give more space to the description of the method, shortening the introduction if needed.
- The sentence “Cardiac left ventricular (LV) segmentation [...] is a necessary step in the processing allowing the identification [...]” should be changed in “Cardiac left ventricular (LV) segmentation [...] allows the identification and diagnosis of cardiac diseases.”
- How were the "best trained proposed models" chosen? This should be clearly stated.

Introduction
- The authors write that “[...] deep learning-based models have been applied for LGE MRI segmentation in various disease areas.” However,  only one paper is cited. A more comprehensive literature review should be provided. For example, the authors could cite [Moccia, Sara, et al. "Development and testing of a deep learning-based strategy for scar segmentation on CMR-LGE images." Magnetic Resonance Materials in Physics, Biology and Medicine 32.2 (2019): 187-195.] and [Li, Lei, et al. "Atrial scar quantification via multi-scale CNN in the graph-cuts framework." Medical Image Analysis 60 (2020): 101595.]
- The benefits of (i) including squeeze and excitation blocks and (ii) processing volumes in a 2.5D fashion should be introduced.
-Why do the authors refer to 2.5D instead of 3D?

Datasets
- How was the manual segmentation performed? Did the clinicians make use of any annotation software? If an expert performed the manual annotation, how was the inter-subject variability computed?

2.5 D proposed RSE-Net Model
- What do the authors mean by special convolutional module?
- I suggest the authors to give more space to the SE module description, considering that introducing this block is the central part of the paper.
- How were the three models chosen? This should be clearly explained in this section.
- The authors should be more accurate in reporting the learning-rate and batch-size values, as well as the used loss function.

Results and Discussion
- Why were the DSC and HD computed in their 2D formulation?
- How was the correlation value computed?
- Table 1 should show also dispersion metrics.
- To my knowledge. the scar tissue is rather well contrasted with respect to the myocardial region in CMR-LGE volumes (that’s why CMR-LGE is used over standard CMR). A figure to show sample challenging slices, with the obtained segmentation, may help the reader in appreciating more the challenges that have to be tackled.
- Please, change "myocardial architecture" with "myocardial anatomy".
- As a general comment, it would be nice to reference publicly available datasets in the field (if any). Testing the proposed methodology on publicly available datasets would promote a fair comparison with the literature.

Minor
- All acronyms should be defined at their first use (e.g., SE)

---

### Official Review · AnonReviewer5 · 2020-03-19
**Method description difficult to follow, but detailed evaluation on sizable dataset**

**Rating:** 3
**Confidence:** 3

**Review:**

This short paper describes a network architecture for segmentation of the left ventricle in short-axis contrast-enhanced MRI scans, aiming specifically at late gadolinium enhancement MRI scans. The authors propose to use three 2D networks with late fusion of the predictions (2.5D). The network architecture makes use of a ResNet50 as feature extractor and uses squeeze-and-excitation blocks to somehow combine the features extracted by the individual residual blocks of the ResNet into a single prediction. Unfortunately, the description of the network architecture is rather hard to follow, and there is also no illustration of the architecture - overall, it is difficult to grasp both the overall structure as well as the main novelty of the architecture.

The propose network was trained and evaluated with a large dataset (~350 scans) and compared with various other architectures. Inter- and intra-observer variations were also measured by repeating the manual segmentation of the test dataset. The method achieves mediocre Dice scores (overall 82% on average), but the paper demonstrates that this performance is close to the inter- and intra-observer agreement.

In summary, the details of the network architecture proposed in this paper are difficult to understand, but even though it is clear that this is still preliminary work that is being presented, the evaluation is quite detailed and the dataset relatively large. Provided that the authors improve the presentation of their method, this could be a good contribution to MIDL.

---

### Official Review · AnonReviewer3 · 2020-03-19
**Segmentation architecture proposed for segmenting LV myocardium in late-gadolinium enhanced cardiac MR images**

**Rating:** 2
**Confidence:** 4

**Review:**

- The authors propose an architecture for segmentation that combines several components that improve over U-net, Segnet and FractalNet baselines. Notable components include squeeze-excite (SE) blocks, residual units, and a 2.5D convolution model.
- The paper is clearly organised, although some important details about the data and architecture are missing.
- Unanswered questions regarding data: What size are the images? For inter-observer variability, what experience did the second annotator have?
- Questions regarding the model: How many layers and filters are there in the model? How many spatial scales/downsampling layers are used? Do you use skip connections? What are the 'specialized convolutions', and how do they improve performance?
- Results on the test data compared to the inter- and intra-observer variability are quite promising. However, no illustrations are shown for qualitative performance and comparison with baseline models, particularly in regions of scar where the authors state models typically struggle.

---

### Meta-Review · Area_Chair1 · 2020-04-05
**MetaReview of Paper135 by AreaChair1**

**Rating:** 3

**Metareview:**

The reviewers have provided detailed comments and listed many valid issues like modest description of previous work, lack of clarity and details regarding the method, and missing details about the reference standard, modest comparison with previous work. However, the addressed problem is clinically relevant, large data set was used which I consider a strength, and for a short paper relatively detailed analysis of the results is presented. If I compare well, the  results achieved here are comparable with those reported on MS-CMRseg challenge. I think this would be a nice contribution to the conference.

**Paper Type:**

both

---

### Decision · Program_Chairs · 2020-04-11

Accept